# Novel Flavonoid Glycosides of Quercetin from Leaves and Flowers of *Gaiadendron punctatum* G.Don. (Violeta de Campo), used by the Saraguro Community in Southern Ecuador, Inhibit α-Glucosidase Enzyme

**DOI:** 10.3390/molecules24234267

**Published:** 2019-11-22

**Authors:** Héctor Cedeño, Sandra Espinosa, José Miguel Andrade, Luis Cartuche, Omar Malagón

**Affiliations:** Departamento de Química, Universidad Técnica Particular de Loja (UTPL), Loja 1101608, Ecuador; hpcedeno@utpl.edu.ec (H.C.); ssespinosa@utpl.edu.ec (S.E.); jmandrade@utpl.edu.ec (J.M.A.); lecartuche@utpl.edu.ec (L.C.)

**Keywords:** *Gaiadendron punctatum*, quercetin glycosides, Loranthaceae, antimicrobial activity, α-glucosidase enzyme inhibitor

## Abstract

*Gaiadandendron punctatum* G.Don. (violeta de campo) is a plant used in traditional medicine by the Saraguro people, an ancient indigenous group that lives in southern Ecuador. From samples collected in the region, six glycoside flavonoids, five with quercetin and one with kaempferol as aglycon, were isolated and characterized from hydroalcoholic extracts of leaves and flowers. Rutin (**2**) was found in flowers and leaves, nicotiflorin (**1**) was found in flowers, artabotryside A (**3**) was found in leaves, and three novel quercetin flavonoid glycosides were isolated, elucidated, and characterized via 1D and 2D NMR experiments (^1^H, ^13^C, COSY, DEPT, HMBC, HSQC, TOCSY, NOESY, ROESY), acid hydrolysis–derivatization–GC-MS analysis, HPLC-MS, IR, UV, and optical rotation. The new quercetin flavonoid glycosides were named hecpatrin (**4**) (isolated from leaves), gaiadendrin (**5**) (isolated from leaves), and puchikrin (**6**) (isolated from flowers). The hydroalcoholic extracts of the leaves presented antimicrobial activity against *Micrococcus luteus*, *Staphylococcus aureus*, and *Enterococcus faecalis* and the hydroalcoholic extract of the flowers was active against *Micrococcus luteus*. However, glycoside flavonoids presented scarce antimicrobial activity against bacteria. Hydroalcoholic extracts from leaves and flowers and their secondary metabolites showed inhibition against the α-glucosidase enzyme at different concentrations. Rutin, gaiadendrin, and nicotiflorin showed competitive α-glucosidase inhibition, while hecpatrin presented non-competitive inhibition.

## 1. Introduction

The Saraguro community is one of the indigenous Kichwa groups of Ecuador, and has settled for centuries in the Loja Province of southern Ecuador. This indigenous community has an ancestral organization and preserves their traditional knowledge, medicine, and culture. For many years, the ethnic groups of Ecuador such as the Saraguros have been known for their use of numerous medicinal plants, and for managing their own healthcare system [1].

The species *Gaiadendron punctatum* G. Don. (violeta de campo) is a Loranthaceae family member [2], known locally as “violeta de campo” in Spanish and “puchik” in the Kichwa language, and is used in traditional preparations by indigenous Saraguro. This community use the aqueous infusion from leaves and flowers as a hair tonic, and in the traditional treatment of bronchitis, hepatic pain, influenza, and strong coughs [3,4]. *G. punctatum* is a parasitic plant [5] located in Central and South America [6], a native shrub or small tree of the Ecuadorean Andes, widely distributed between 1500 and 4000 m.a.s.l. [7,8], and common in tropical rain forests, subalpine areas, and overgrown bushes [9]. In southern Ecuador, is has been located in different sites, such as Cajanuma and Saraguro, among others [10]. The Loranthaceae family has been reported to have anti-inflammatory [11], antioxidant [12,13], antimalarial [14], antimicrobial, antiviral [15], and gastrointestinal properties [16]. Some studies performed in this family have reported the presence of glycosylated flavonoids [17] and cardiac glucosides [18]. In general, flavonoids and their glycosides have been shown to have a wide range of biological activity, including antioxidant [19,20], antimicrobial [21], vasodilatory [22], inhibitory enzymatic [23], anti-inflammatory [24], anti-tumor [25], and cytotoxic effects [26].

The interest in this plant arises from the few biological [27] and phytochemical studies that have been performed and its extensive medicinal use to treat postpartum conditions, coughs [28], nerve conditions, insomnia [29], problems related to puberty, smallpox, and measles [30]. The aim of this study was to separately isolate the relevant compounds from hydroalcoholic extracts from leaves and flowers of *G. punctatum*, and to elucidate them through 1D and 2D NMR experiments (1H, 13C, COSY, HMBC, HSQC, TOCSY, NOESY, ROESY), hydrolysis of glycosides and GC-MS analysis, and subsequent characterization using HPLC-MS, IR, UV and optical rotation. Moreover, we evaluated the antimicrobial activity and inhibition against α-glucosidase of the hydroalcoholic extracts and some of the isolated molecules.

## 2. Results

### 2.1. Isolation of Compounds

One kaempferol glycoside (nicotiflorin (**1**)) and five quercetin glycosides (rutin (**2**), artabotryside A (**3**), hecpatrin (**4**), gaiadendrin (**5**), and puchikrin (**6**)) were isolated from hydroalcoholic extracts of the leaves and flowers of *G. punctatum* (Figure 1). Compounds **4**–**6** had not been reported previously in the literature, based on the data acquired in our study, while compounds **1**–**3** were known flavonoid glycoside structures. Both the 1D and 2D NMR analysis and comparison with literature allowed the elucidation of the absolute structures of compounds **1**–**6**. Additional information about the spectral analysis of compounds **1**–**6** can be found in the Appendix A.

### 2.2. Characterization of Isolated Compounds

Compound **1** was found in flowers as an amorphous yellow substance, with a 6.3 mg yield from 1.1 grams of hydroalcoholic extract. NMR spectrum: ^1^H NMR (400 MHz, methanol-*d*_4_) δ 8.07 (d, *J* = 8.9 Hz, 1H), 6.89 (d, *J* = 8.9 Hz, 2H), 6.41 (d, *J* = 2.1 Hz, 1H), 6.21 (d, *J* = 2.1 Hz, 1H), 5.13 (d, *J* = 7.5 Hz, 1H), 4.52 (d, *J* = 1.7 Hz, 1H), 3.81 (dd, *J* = 10.9, 1.4 Hz, 1H), 3.63 (dd, *J* = 3.4, 1.7 Hz, 1H), 3.52 (dd, *J* = 9.5, 3.4 Hz, 1H), 3.48 – 3.44 (m, 1H), 3.43 (t, *J* = 1.9 Hz, 1H), 3.41 (d, *J* = 2.7 Hz, 1H), 3.39 – 3.34 (m, 1H), 3.33 (d, *J* = 1.6 Hz, 0H), 3.28 (d, *J* = 2.9 Hz, 1H), 3.25 (d, *J* = 2.3 Hz, 1H), 1.12 (d, *J* = 6.2 Hz, 3H). ^13^C NMR (101 MHz, methanol-*d*_4_) δ 17.92, 68.57, 69.74, 71.45, 72.10, 72.30, 73.90, 75.76, 77.23, 78.15, 94.99, 100.08, 102.44, 104.62, 105.58, 116.15, 122.76, 132.37, 135.50, 158.60, 159.39, 161.53, 163.01, 166.40, 179.40. The NMR spectrum was compared with the literature [31], corresponding to nicotiflorin. COSY (S4) and HMBC correlations (S6) confirmed the identity of the compound (Figure 2). Nicotiflorin possesses the molecular formula C_27_H_30_O_15_
*(*calc. 594.15847 Da*)* and [α]^23^_d_ = +0.34 (C = 0.34 mg/mL, MeOH), IR (cm^−1^) absorption bands were present at 3317, 2944, 2831, 1449, 1110, 1023, 611, 527, and 517. LC-ESI-MS analysis showed a signal at m/z [M+H+Na]^+^ 617.78.

Compound **2** was found in leaf and flower extracts as a yellow amorphous powder. The yield was 6.3 mg from 2.4 g of leaf hydroalcoholic extract and 19.8 mg from 1.1 g of flower hydroalcoholic extract. NMR spectrum: ^1^H NMR (400 MHz, methanol-*d*_4_) δ 7.67 (d, *J* = 2.1 Hz, 1H), 7.63 (dd, *J* = 8.4, 2.2 Hz, 1H), 6.88 (d, *J* = 8.4 Hz, 1H), 6.41 (d, *J* = 2.1 Hz, 1H), 6.22 (d, *J* = 2.1 Hz, 1H), 5.11 (d, *J* = 7.6 Hz, 1H), 4.52 (d, *J* = 1.7 Hz, 1H), 1.12 (d, *J* = 6.2 Hz, 3H). ^13^C NMR (101 MHz, methanol-*d*_4_) δ 17.88, 48.36, 48.58, 48.79, 49.00, 49.07, 49.21, 49.29, 49.43, 49.64, 68.55, 69.71, 71.40, 72.11, 72.24, 73.93, 75.73, 77.24, 78.19, 94.86, 99.95, 102.43, 104.70, 105.64, 116.06, 117.68, 123.13, 123.55, 135.62, 145.86, 158.53, 159.35, 163.02, 166.06, 179.44. The ^1^H and ^13^C NMR spectra were compared with the literature [32], confirming the presence of rutin. COSY (S12) and HMBC correlations (S14) confirmed the identity of the compound (Figure 3).

The molecular formula of rutin is C_27_H_30_O_16_
*(*calc. 610.153390 Da*)*. [α]^23^_d_ = +0.07 (C = 0.27 mg/mL, MeOH). The IR (cm^−1^) absorption bands were present at 3330, 2945, 2831, 1449, 1118, 1024, 629, 556, 532, and 515. LC-ESI-MS analysis showed a signal at m/z [M+Na]^+^ 633.17.

Compound **3** was isolated from hydroalcoholic leaves extract as an amorphous yellow substance, with a 40.6 mg yield from 2.1 g of hydroalcoholic extract. NMR spectrum: ^1^H NMR (400 MHz, methanol-d4) δ 7.62 (d, J = 2.2 Hz, 1H), 7.56 (dd, J = 8.4, 2.2 Hz, 1H), 6.89 (d, J = 8.4 Hz, 1H), 6.38 (d, J = 2.1 Hz, 1H), 6.19 (d, J = 2.1 Hz, 1H), 5.53 (d, J = 5.2 Hz, 1H), 5.09 (d, J = 1.7 Hz, 1H), 4.10 (dd, J = 6.9, 5.2 Hz, 1H), 3.91 (dd, J = 3.4, 1.7 Hz, 1H), 3.89–3.84 (m, 1H), 3.83 – 3.79 (m, 2H), 3.70 (dd, J = 9.6, 3.4 Hz, 1H), 3.66–3.59 (m, 1H), 3.41–3.36 (m, 1H), 3.35 (d, J = 1.5 Hz, 1H), 1.06 (d, J = 6.2 Hz, 3H). ^13^C NMR (101 MHz, cd3od) δ 17.6041, 65.3038, 68.4979, 70.1392, 72.2519, 72.4240, 72.8500, 73.9956, 77.1973, 94.6156, 99.7926, 101.1583, 102.2285, 105.7600, 116.2436, 117.1521, 123.1049, 123.1472, 135.1470, 146.1064, 149.7710, 158.3918, 158.5182, 163.1654, 165.9655, 179.4151.

^1^H and ^13^C NMR spectra were compared with the literature [33], confirming the presence of artabotryside A. COSY (S19) and HMBC correlations (S21) confirmed the identity of the compound (Figure 4). The molecular formula of artabotryside A is C_26_H_28_O_15_*(*calc. 580.14282 Da), [α]^23^_d_ = -69.99 (C = 0.15 mg/mL, MeOH). The UV spectrum (CH_3_OH) λ_max_(log ɛ) at 247 (5.18) nm showed chromophore peaks (242, 247, 260, 267, and 272 nm). The IR (cm^−1^) absorption bands were present at 3311, 2944, 2831, 1449, 1116, 1021, 1021, 617, 575, and 502.

Compound **4** was isolated from leaf extracts as an amorphous yellow substance and yielded 9.3 mg from 2.4 grams of hydroalcoholic extract. The spectral NMR details are given in Table 1. The presence of the quercetin aglycon was confirmed by the chemical shift of protons 6 (6.18 ppm), 8 (6.37 ppm), 2’ (7.62 ppm), 5’ (6.87 ppm), and 6’ (7.60 ppm). The glycoside region analysis showed two anomeric protons in 1’’ (d, 5.75 ppm) and 1’’’ (d, 5.23 ppm). Moreover, a methyl signal as a doublet was found at 0.97 ppm, which is usual in rhamnose. COSY and HMBC were able to find the characteristic spin system of sugars. Anomeric proton 1’’ was correlated with 2’’ (d, 3.66 ppm), proton 2’’ correlated with proton 3’’ (d, 3.36 ppm), 3’’ associated with proton 4’’ (m, 3.26–3.19 ppm), 4’’ correlated with 5’’ (d, 3.56 ppm), and finally, 5’’ associated with 6’’(2H, d 3.85ppm, d 3.72ppm). According to the HMBC data, this sugar O-links through the 6’’ carbon with C 4’. Coupling constants confirmed this result. This behavior is typical for β-glucose.

Proton 1’’ presented an axial–axial coupling constant (J = 7.7 Hz) with proton 2’’, 2’’ showed an axial–axial coupling constant (J = 9.3 Hz) with proton 3’’, proton 3’’ exhibited an axial–axial coupling constant (J = 9.3 Hz) with proton 4’’, proton 4’’ displayed a multiplet signal that revealed an axial–axial correlation with 5’’, proton 5’’ presented a triple doublet (J = 8.9, 1.7Hz), proton 6’’ (2H) showed a doublet (J = 2.2Hz) at 3.72 ppm and a multiplet at 3.55 ppm. Selective TOCSY analysis determined the precise assignment of proton multiplicity in the sugars (S32). HMBC data showed a correlation between the anomeric proton of rhamnose (1’’’) with the 2’’carbon of glucose. Bibliographic revision of the chemical shift confirmed that the bonds between the aglycone and the first sugar and between the first and second sugar were in an α configuration [34]. Additionally, compound **4** was acid hydrolyzed. The glycoside fraction was derivatized and analyzed via GC-MS, confirming the presence of α-glucose and α-rhamnose in the structure of the glycoside flavonoid (S33). This structure has not been previously reported in literature and corresponds to a new natural product, quercetin-4’-O-(2-O-β-rhamno-pyranosyl-6-β-glucopyranoside) (Figure 5). Compound **4**, as a new natural product, was named hecpatrin (**4**).

Hecpatrin (**4**) possesses the molecular formula C_27_H_30_O_16_
*(*calc. 610.1533 Da). [α]^23^_d_ = + 0.19 (C = 0.65 mg/mL, MeOH). The degree of unsaturation of hecpatrin is 13, which corresponds to eight unsaturated bonds to aromatic rings A and B, one double bond in flavone ring C, a carbonyl group at carbon 4, one ether bond with the flavone, and two ether bonds, one for each sugar. Its UV spectrum (CH_3_OH)λ_max_(log ɛ) at 261 (3.99) nm showed chromophore peaks (261, 267, and 351 nm). The IR (cm^−1^) absorption bands were at 3326, 2943, 2831, 1449, 1111, 1023, 642, 567, 518, and 502.

Compound **5** was isolated from leaf extract as an amorphous yellow substance and yielded 51.1 mg from 2.4 grams of hydroalcoholic extract. The spectral NMR details are given in Table 1. The presence of the quercetin aglycon was confirmed by the chemical shift of protons 6 (6.18 ppm), 8 (6.37 ppm), 2’ and 6’ (7.62–7.59 ppm), and 5’ (6.87 ppm). The glycoside region analysis showed three anomeric protons: 1’’ (d, 5.59 ppm), 1’’’ (d, 4.51ppm), and 1’’’’ (d, 5.23 ppm). Moreover, two methyl signals as a doublet were found at 1.08 ppm and 1.01 ppm, which is usual in rhamnose. COSY and HMBC were able to identify the characteristic spin system of sugars. Anomeric proton 1’’ was correlated with 2’’ (dd, 3.65 ppm), proton 2’’ correlated with proton 3’’ (dd, 3.50 ppm), 3’’ associated with proton 4’’ (q, 3.33 ppm), 4’’ correlated with 5’’ (m, 3.29–3.26 ppm), and finally, 5’’ associated with 6’’(2H, d 3.84 ppm, d 3.39 ppm). According to HMBC data, this sugar O-links through C1’’ and C3. Coupling constants confirmed this result. This behavior is typical for β-glucose or α-mannose.

Proton 1’’ presented an axial–axial coupling constant (J = 7.6 Hz) with proton 2’’, 2’’ showed an axial–axial coupling constant (J = 9.3 Hz) with proton 3’’, proton 3’’ exhibited an axial–axial coupling constant (J = 3.4 Hz) with proton 4’’, proton 4’’ displayed a non-identified coupling constant (m) with 5’’, proton 5’’ presented a multiplet, protons at 6’’ (2H) showed a doublet at 3.84 ppm (J = 1.4 Hz) and a doublet at 3.39 ppm (J = 6.5Hz). The selective TOCSY analysis determined the precise assignment of proton multiplicity in the sugars (S44). HMBC data showed a correlation between the anomeric protons of rhamnose (1’’’ and 1’’’’) with the 2’’ and 6’’ carbons of the other sugar. Compound **5** was acid hydrolyzed. The glycoside fraction was derivatized and analyzed via GC-MS, confirming the presence of β-glucose and β-rhamnose in the structure of the glycoside flavonoid (S45). Bibliographic revision of the chemical shift confirmed that the bonds between the aglycone and the first sugar and between glucose and rhamnose 1 bonded to C2’’ were in β configurations, and the one between mannose and rhamnose linked to C6’’ was β [34]. This structure has not been previously reported in the literature and corresponds to a new natural product, quercetin-3-O-(2,6-di-O-β,β-rhamno-pyranosyl-β-glucopyranoside) (Figure 6). The 2D ROESY experiment showed a relationship between protons 1’’’’–5’’’’, 1’’’’–2’’, 1’’’–6’’, and 1’’’–5’’’. It confirmed the proposed absolute configuration for the glycoside. Compound **5**, as a new natural product, was named gaiadendrin (**5**). Gaiadendrin (**5**) possesses the molecular formula C_33_H_40_O_30_ (calc. 756.1129 Da). The degree of unsaturation of gaiadendrin is 14, corresponding to eight unsaturated bonds to aromatic rings A and B, one double bond in flavone ring C, a carbonyl group in carbon 4, one ether bond with the flavone, and three ether bonds, one for each sugar. [α]^23^_d_ = +0.05 (C = 5.88 mg/mL, MeOH). The UV spectrum (CH_3_OH)λ_max_(log ɛ) at 257 (3.54) nm showed chromophore peaks (257, 261, 268, 294, and 355 nm). The IR (cm^−1^) absorption bands were at 3319, 2943, 2832, 1450, 1111, 1025, 640, 570, 536, 526, and 511. LC-ESI-MS analysis showed a signal at *m/z* ([M+Na]^+^ 779.34.

Compound **6** was isolated from flower extract as an amorphous yellow substance, and yielded 12.7 mg from 1.2 grams of hydroalcoholic extract. The spectral NMR details are given in Table 1. The ^1^H and ^13^C results were similar to those for gaiadendrin (**5**), with the exception of one proton behavior in the 3.5–3.8 ppm region. This spectroscopic information allowed the inference of slight differences between compound **6** and gaiadendrin (S58–S60). The presence of the quercetin aglycon was confirmed by the chemical shift of protons 6 (6.19 ppm), 8 (6.37 ppm), 2’ and 6’ (7.64–7.57 ppm), and 5’ (6.87 ppm). The glycoside region analysis showed three anomeric protons at 1’’ (d, 5.59 ppm), 1’’’ (d, 4.51ppm), and 1’’’’ (d, 5.22 ppm). Moreover, two methyl signals as doublets were found at 1.08 ppm and 1.00 ppm, which is usual in rhamnose. COSY and HMBC were able to identify the characteristic spin system of sugars. Anomeric proton 1’’ was correlated with 2’’ (dd, 3.64 ppm), proton 2’’ correlated with proton 3’’ (dd, 3.49 ppm), 3’’ associated with proton 4’’ (m, 3.34–3.32 ppm), 4’’ correlated with 5’’ (m, 3.3–3.26 ppm), and finally, 5’’ associated with 6’’(2H, d 3.83 ppm, d 3.39 ppm). According to the HMBC data, this sugar O-links through the 1’’ carbon with C3. Coupling constants confirmed this result. This behavior is typical for β-glucose or α-mannose.

Proton 1’’ presented an axial–axial coupling constant (J = 7.7 Hz) with proton 2’’, 2’’ showed an axial–axial coupling constant (J = 9.2 Hz) with proton 3’’, proton 3’’ exhibited an axial–axial coupling constant (J = 9.5 Hz) with proton 4’’, proton 4’’ displayed a multiplet signal that revealsed a non-identified coupling constant with 5’’, proton 5’’ presented a multiplet, protons at 6’’ (2H) showed a doublet at 3.83 ppm (J = 1.5Hz) and a multiplet at 3.39 ppm. Selective TOCSY analysis determined the precise assignment of proton multiplicity in the sugars (S56). HMBC data showed a correlation between the anomeric protons of rhamnoses (1’’’ and 1’’’’) with the C2’’ and C6’’ of the other sugar. Compound **6** was acid hydrolyzed. The glycoside region was derivatized and analyzed via GC-MS, confirming the presence of α-mannose and β-rhamnose in the structure of the glycoside flavonoid (S57). Bibliographic revision of the chemical shift confirmed that the bonds between the aglycone and the first sugar (α-mannose) and between mannose and rhamnose linked in carbon 2’’ were in β configuration, and the one between mannose and rhamnose linked to C’’ was β [34]. This structure has not been previously reported in literature and corresponds to a new natural product, quercetin-3-O-(2,6-di-O-β,β-rhamno-pyranosyl-α-mannopyranoside) (Figure 7). The 2D-ROESY experiment confirmed a relationship between protons 1’’’’–2’’’’ and between protons 3’’’’–2’’’’. There was no ROE between 1’’’’ and 2’’, which confirmed the proposed configuration. Compound **6**, as a new natural product, was named puchikrin (**6**)**.** Puchikrin (**6**) possesses the molecular formula C_33_H_40_O_30_
*(*calc. 756.1129 Da). The degree of unsaturation of puchikrin is 14, which corresponds to eight unsaturated bonds to aromatic rings A and B, one double bond in flavone ring C, a carbonyl group in carbon 4, one ether bond with the flavone, and three ether bonds, one for each sugar. [α]^23^_d_ = + 0.11 (C = 0.85 mg/mL, MeOH). The UV spectrum (CH_3_OH)λ_max_(log ɛ) at 256 (3.93) nm showed chromophore peaks (256, 268, 295, and 355 nm). The IR (cm^−1^) absorption bands were at 3310, 2943, 2832, 1449, 1117, 1023, 630, 591, 557, 536, 515, and 506. The LC-ESI-MS analysis showed a signal at m/z ([M+H+Na]^+^) 780.52.

The spectral NMR details are given in Table 1.

### 2.3. Analysis of Antimicrobial Activity

The antimicrobial activity results are shown in Table 2.

### 2.4. Inhibitory Activity of the α-Glucosidase Enzyme

The enzymatic inhibitory activity results of the hydroalcoholic extracts and selected compounds are shown in Table 3.

#### Kinetic Analysis of Pure Compounds

The kinetics of enzymatic inhibitory activity results are presented in Figure 8 and Table 4.

## 3. Discussion

The 1D and 2D NMR experiments and hydrolysis–GC-MS analysis of glycosides determined that compounds hecpatrin (**4**)**,** gaiadendrin (**5**), and puchikrin (**6**) had not previously been reported, while nicotiflorin (**1**), rutin (**2**), and artabotryside A (**3**) had already been reported, according to bibliographic comparisons [33,35,36,37].

According to the results shown in Table 2, only the hydroalcoholic extract of leaves exerted a moderate [37] effect against *M. luteus*, and it was non-active against all the bacteria and fungi tested. None of the isolated compounds displayed any antimicrobial effect at the maximum dose tested (range 200 to 340 μM).

Both the extracts and the isolated compounds showed enzymatic inhibitory activity comparable or superior to the current medication used in the treatment of diabetes mellitus (acarbose) [38], and the kinetics of the active compounds were examined (hecpatrin, rutin, gaiadendrin, and nicotiflorin). Rutin and nicotiflorin exhibited inhibitory activity similarly to in previous reports [39,40,41,42], while compounds hecpatrin (KI = 10.86) and gaiadendrin (KI = 15.35) exhibited a higher inhibitory activity compared to the two aforementioned compounds.

## 4. Materials and Methods

### 4.1. General Information

The dehydration of the vegetable samples was carried out in a drying room in trays at 40 °C and 35% humidity. The maceration was performed in an aluminum tank with a propeller engine. The chlorophyll extraction process was executed in a 10 g solid phase extraction (SPE) column, using 100 mL MeOH:H_2_O (8:2) as the mobile phase. The drying of the extracts, fractions, and compounds was accomplished in a BUCHI R300 rotary evaporator at 175 rpm, 35 °C, with an absolute pressure of 20 to 150 mBar. In addition, a Labconco Corporation model 7754047 lyophilizer was used in freeze-drying the extracts.

The fractionation and purification of the samples were carried out using a Reveleris PREP Purification System, using prefabricated cartridges of 12 g of silica SG_60_ or silica RP-18 from Reveleris® with a flow of 30 mL/min, with an evaporative light scattering detector (ELSD) and a UV light absorption detector. Glass columns of 1.8 cm in diameter and 70 cm in height were also used, using 10 g of sephadex resin LH-20 as the stationary phase and methanol as the mobile phase. Finally, a preparative Thin Layer Chromatography technique (TLC) was used with silica gel TLC plates F_254_ of 10 × 20 cm, using as a mobile phase a mixture of AcOEt:MeOH:H_2_O (8:1.2:0.8).

The sample elucidation process was performed based on the nuclear magnetic resonance (NMR) spectra of ^1^H, ^13^C, DEPT, COSY, HMBC, and HSQC at 400 MHz and 100 MHz, using a VARIAN Agilent 400 MHz Premium Shielded with a MR Y 0021953 magnet and with a MR1005 W031 console and ONE NMR Probe 5 mm. Selective TOCSY, ROESY, and NOESY experiments were performed in a Bruker Avance Neo Ascend 500 MHz with a 1.7 mm microprobe. Both pieces of equipment were located at the Chemistry and Exact Sciences Department at UTPL.

The mass spectra were obtained by direct injection into the Bruker amaZon speed electrospray ionization equipment (ESI), with nitrogen being the ionization gas. The infrared spectra (FTIR) were measured using Nicolet iS 10 equipment (Thermo Scientific), and using the OMNIC32 program to visualize the spectra. Finally, the analysis of optical rotation was carried out in a Mrc-Automatic Polarimeter P810 in a cell of 1 dm length. The samples used in the determination of the mass spectra (EM), infrared spectra (FTIR), and optical rotation [α]^d^_25_ were dissolved in HPLC-grade methanol from MERCK.

The analysis of enzymatic inhibitory activity was performed on an Epoch™ 2 Microplate Spectrophotometer, using 96 well microplates at a wavelength of 405 nm for 60 minutes. The antimicrobial assay was done in a Yamato brand incubator model IC 800 at 37 °C for 1 day.

The bidistilled technical grade solvents used in the elucidation of the chromatographic columns and TLC plates were hexane (hex) from Philips 66 Specialty Solvents, ethyl acetate (AcOEt) from Celanes, methanol from Solventis® (MeOH), and distilled water (H_2_O), while the reagents used in the spectroscopic analyses were deuterated methanol (CD_3_OD) from Aldrich, code: 15947-100 G, and HPLC-grade methanol from Merck. TLC plates were developed with alcohol solutions of 5% sulfuric acid (H_2_SO_4_) reagent grade from Merck and 1% vanillin from Merck.

The reagents used in the analysis of the biological activity were phosphate-buffered saline (PBS) (Sigma, code: 101531808), dimethylsulphoxide (DMSO) ( Fisher Chemical, code: 10127403), BBL^TM^ TrypticaseTM SoY Broth (Becton Dickinson and Company, ref: 211768), DifcoTM nutrient broth (Becton Dickinson and Company, ref: 234000), Müeller–Hinton broth (Becton Dickinson and Company, ref: 275730), DifcoTM sabouroud dextrose broth (Becton Dickinson and Company, ref: 238230), BBLTM brain–heart infusion (Becton Dickinson and Company, ref: 211059), 4-nitriphenyl α-D-glucospyranoside (Sigma, code: N1377-56), *Saccharomyces cerevisiae* α-glucosidase (Sigma, code: G5003-1KV), and acarbose (Sigma, code: A8980-1G).

### 4.2. Collection of Plant Material and Obtaining Extracts

Samples of leaves and flowers were collected from the plant species *Gaiadendron punctatum* G.Don. (violeta de campo) in the Zhindar neighborhood of the canton of Saraguro (3°36’6’’ S and 79°16’2’’ W, at 2700 masl) in Loja Province, Ecuador with the research permit MAE-DBN-CM-2016-0048, coded with voucher PPN-lo-001, and finally deposited in the Herbarium of the Universidad Técnica Particular de Loja. 

The leaves and flowers were crushed and macerated in different containers, with hexane solvents and the ethanol:water solution (7:3).

### 4.3. Dechlorophyllation and Fractionation of Hydroalcoholic Extracts of Leaves and Flowers

From 10.4 g of the leaf hydroalcoholic extract, 8.26 g of dechlorophylled extract was obtained using solid phase extraction (SPE) and the solvent mixture MeOH:H_2_O (8:2). This extract was fractionated with a gradient of AcOEt:MeOH:H_2_O (100:0:0) to AcOEt:MeOH:H_2_O (80:12:8) in SG_60_. From leaf extract two fractions were obtained and coded: **E.H.7** (235.6 mg) and **E.H.10** (88.6 mg). **E.H.7** was fractionated using a gradient of ACOEt:MeOH:H_2_O (100:0:0) to ACOEt:MeOH:H_2_O (80:12:8) in SG_60_, to give the fraction coded **E.H.7.4** (72.7 mg).

For the flower extract, 1.1 g was fractionated in RP-18 silica with a gradient from MeOH:H_2_O (5:95) to MeOH:H_2_O (100:0), obtaining the main fractions coded **E.F.4**, **E.F.6**, and **E.F.7** (482.9 mg). The fractions **E.F.4**, **E.F.6**, and **E.F.7** were fractionated in RP-18 silica using MeOH:H_2_O (5:95) to MeOH:H_2_O (100:0) as the mobile phase, resulting in fractions **E.F.4**, **E.F.6**, and **E.F.7.5** (45.3 mg), **E.F.4**, **E.F.6**, and **E.F.7.7** (73.7 mg) and **E.F.4**, **E.F.6**, and **E.F.7.10** (18.2 mg).

### 4.4. Purification of Secondary Metabolites

From the hydroalcoholic leaf extract without chlorophyll, four compounds were isolated: rutin (**2**), artabotryside A (**3**), hecpatrin (**4**), and gaiadendrin (**5**). Artabotryside A was obtained via elution in RP_18_ silica with MeOH:H_2_O (8:2) as the mobile phase. **E.H.10** fraction was eluted in MeOH:H_2_O (5:95) to MeOH:H_2_O (100:0), in RP-18 silica, resulting in the **E.H.10.2** fraction elucidated as gaiadendrin (**5**). Hecpatrin (**4**) and rutin (**2**) were obtained from the elution of **E.H.7.4.** in MeOH:H_2_O (5:95) to MeOH:H_2_O (100:0) in RP-18 silica.

From the hydroalcoholic extract of the flowers, nicotiflorin (**1**), rutin (**2**), and puchikrin (**6**) were isolated. Rutin was obtained by fractionating the **E.F.4**, **E.F.6**, **E.F.7.7** sample in a sephadex LH-20 column using MeOH as the mobile phase. Nicotiflorin was isolated from a preparative TLC of fraction **E.F.4**, **E.F.6**, **E.F.7**. Puchikrin was isolated from the **E.F.4**, **E.F.6**, **E.F.7.5** fraction in a sephadex LH-20 column using MeOH as the mobile phase.

### 4.5. Characterization of the Obtained Metabolites

The obtained metabolites were characterized using the analytical techniques of nuclear magnetic resonance (NMR), mass spectrometry (MS), infrared spectrometry with Fourier transform (IRTF), UV spectrometry, and determination of specific rotation [α]_d_.

### 4.6. Hydrolysis and Derivatization of the New Compounds

The derivatization process was carried out via the use of 37% hydrochloric acid (Pharmco), using vials (0.93 mL) of NMethyl-N-trimetylsily trifluoroacetamide (MSTFA) from Restek, code: 35.600, and hydroxylamine hydrochloride (T. Baker). The gas chromatography equipment used was an Agilent Technologies gas chromatograph 6890N and a quadrupole mass spectrometry detector 5973 (Santa Clara, CA, USA) operating in SCAN mode with electronic ionization (70 eV). The gas chromatograph was equipped with a non-polar stationary phase capillary column DB-5MS (Agilent Technologies) (5% phenyl-methylpolysiloxane, 30 m, 0.25 mm internal diameter, and 0.25 µm film thickness; J & W Scientific, Folsom, CA, USA) located at the Chemistry and Exact Sciences Department at UTPL. The following operative conditions were applied for the GC-MS analysis with the DB-5MS column: He was the carrier gas, with a flow rate of 1 mL/min; the injection volume was 1 µL. The injector was operated in a split ratio of 40:1. The injection temperature was set at 280 °C. The elution was conducted according to the following temperature program: from 100 °C to 180 °C at a rate of 10 °C/min, with the third gradient to 240 °C a rate of 25 °C/min. At the end, the oven temperature was kept at 240 °C for 10 minutes. D-mannose standard was obtained from Larix and D(+)-glucose from Merck.

The hydrolysis of hecpatrin, gaiadendrin, and puchikrin began with the preparation of 1M HCl and the use of reflux equipment for 1 hour, using 16 mL of the aforementioned solution and 1 mg of sample. After the hydrolysis process, separation of flavonol and sugars was carried out by means of a separatory funnel, using ethyl acetate and distilled water. The aqueous phase was dried by evaporation under vacuum. In the case of hecpatrin and puchikrin, the sample was placed in a 200 µL insert and added to 100 µL of derivatizer. For gaiadendrin, 10 µL of a 50 mg/mL solution of hydroxylamine hydrochloride in pyridine was added to the sample in a 200 µL insert and placed in an oven at 80 °C for 30 minutes. Finally, 100 µL of derivatizer was added to the sample. The reaction lasted approximately 1 hour, was optimized using a vortex device for 5 minutes, and was finally injected into the GC-MS equipment. The method and analysis of results were based on selected references [43,44,45].

### 4.7. Biological Activity

#### 4.7.1. Antimicrobial Activity

Antimicrobial activity was evaluated against one Gram-negative and three Gram-positive bacteria and one yeast, as shown in Table 2. All strains were incubated overnight at 37 °C in Müeller–Hinton (MH) broth for bacteria and Sabouraud for yeasts.

Minimum inhibitory concentration (MIC) was determined by two-fold serial dilution method using 96 well microtiter plates [46]. Twenty microliters of a solution of each extract (80 mg/mL) or compounds (approximately 1.6 mM) in DMSO:H_2_O (1:1 ratio) was mixed with 180 µL of Müeller–Hinton Broth or Sabouraud broth (for *C. albicans*), and 100 µL of this starting solution was placed on the second row of the plate, containing 100 µL of medium. The procedure was continued until the last row of the plate, where 100 µL was finally discarded. Next, 100 µL of bacterial or fungal inoculum was placed on each dilution to reach a final concentration of 1 × 105 cfu/mL, and sample concentrations ranging from 4000 µg/mL to 31.25 µg/mL for extracts and 320 µM to 2.5 µM for the compounds. Gentamicin (1 mg/mL) was used for Gram-negative bacteria, penicillin (0.25 mg/mL) was used for Gram-positive bacteria and anfotericin B (0.2 mg/mL) for *Candida albicans*. The microplates were sealed and incubated at 37 °C for 24 hr.

#### 4.7.2. α-Glucosidase Inhibition Assay

The inhibitory effect against α-glucosidase enzyme was measured according to the method described by Tao et al. [47], with slight modifications. p-Nitrophenyl-α-D-glucopyranoside (PNPG) was used as substrate, and the enzyme reaction was carried out in a 96 well microtiter plate and recorded on a microplate spectrometer. The extracts were dissolved in a mixture of DMSO andH_2_O (1:1 ratio) at a final concentration of 3 mg/mL. The pure compounds were dissolved in the same solvent mentioned above to ensure complete solubility. The maximum concentration tested for a given compound was 280 µM. Several dilutions in PBS for extracts and compounds were made when complete enzyme inhibition was reached. First, 75 μL of PBS (SIGMA-P4417) was mixed with 5 μL of the sample (extract or compound solutions) and 20 μL of the enzyme solution (0.15 U/mL in PBS pH 7.4); the mixture was then pre-incubated at 37 °C for 5 min prior to the initiation of the reaction by adding the substrate. After pre-incubation, 20 μL of PNPG (5 mM in phosphate buffer, pH 7.4) was added before incubation at 37 °C. The amount of p-nitrophenol (p-NP) released was monitored in an EPOCH 2 (BIOTEK®) microplate reader at 405 nm at 5 min intervals for 60 min. Acarbose was used as a positive control. IC_50_ values were determined by non-regression analysis, log[inhibitor] vs normalized response–variable slope [48] model, using GraphPad Prism 8.0.1.

#### 4.7.3. Kinetic Analysis of Pure Compounds

Km and Vmax were determined for the active isolated flavonoids except puchikrin, which exhibited weak inhibition, by measuring the change in enzyme velocity at different concentrations of PNPG and different concentrations of inhibitor [49,50]. Progress curves for α-glucosidase with 1, 2.5, 5, 10, and 15 mM initial concentrations of PNPG were made and, once the type of inhibition had been determined, Ki was calculated. The amount of enzyme employed was the same used in the enzyme inhibition assay described above, and the same procedure was followed. All the results were analyzed on GraphPad Prism 8.0.1 and are expressed as the mean of three replicates with a 95% CI.

## Figures and Tables

**Figure 1 molecules-24-04267-f001:**
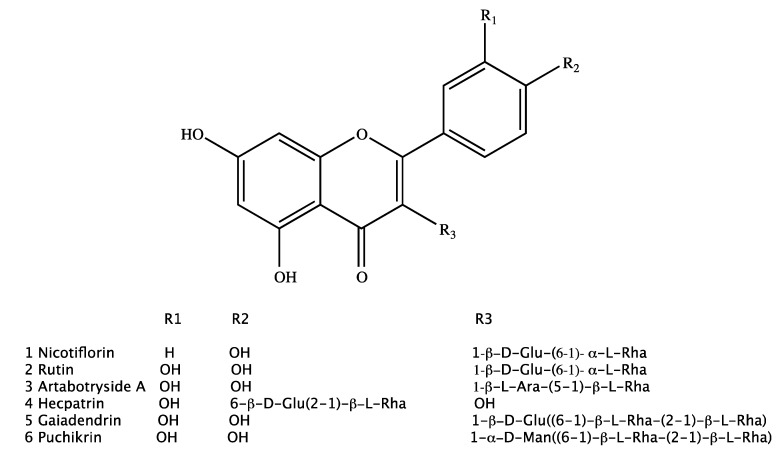
Compounds **1** to **6** isolated from *Gaiadendron punctatum.*

**Figure 2 molecules-24-04267-f002:**
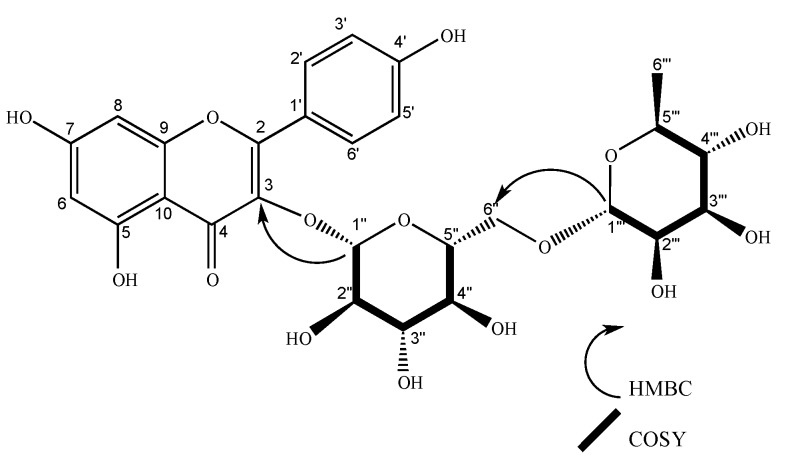
Structure of nicotiflorin (**1**). COSY and HMBC key correlations are shown for glycosidic units.

**Figure 3 molecules-24-04267-f003:**
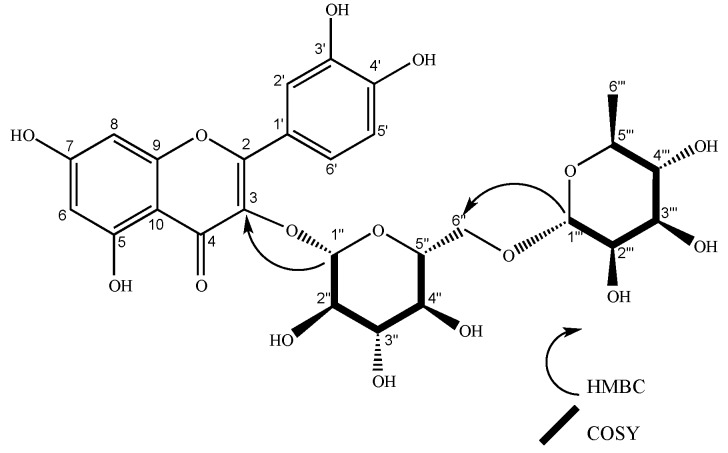
Structure of rutin (**2**). COSY and HMBC key correlations are shown for glycosidic units.

**Figure 4 molecules-24-04267-f004:**
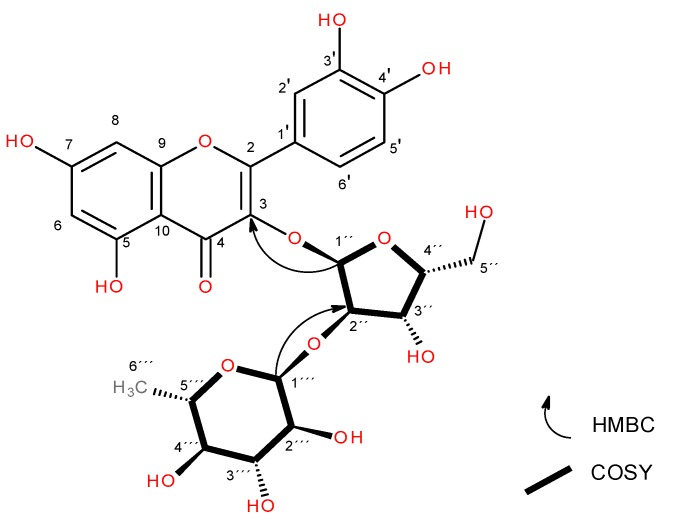
Structure of artabotryside A (**1**). HMBC key correlations are shown for glycosidic units.

**Figure 5 molecules-24-04267-f005:**
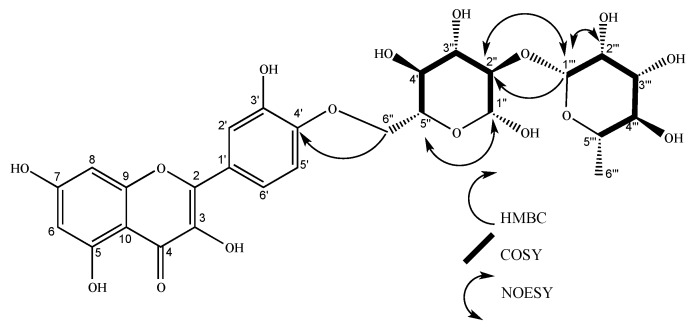
Structure of hecpatrin. COSY and HMBC key correlations are shown for glycosidic units.

**Figure 6 molecules-24-04267-f006:**
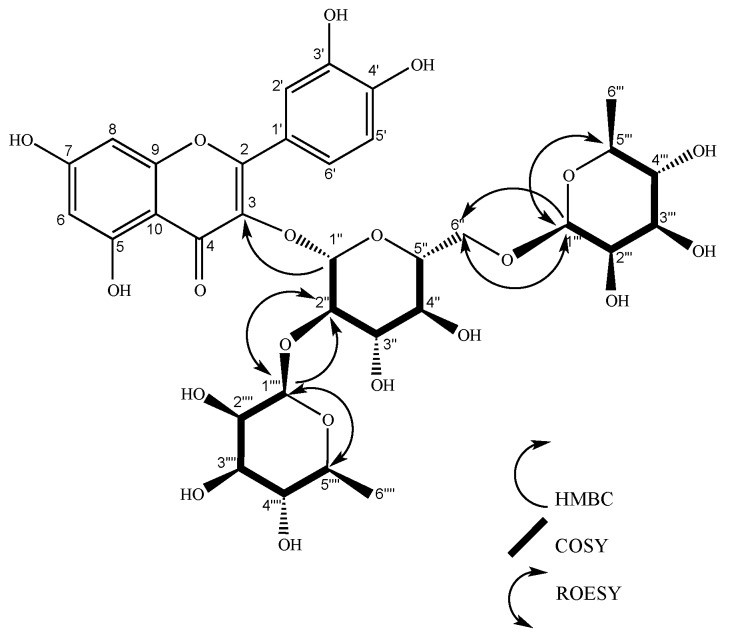
Structure of gaiadendrin. COSY and HMBC key correlations are shown for glycosidic units.

**Figure 7 molecules-24-04267-f007:**
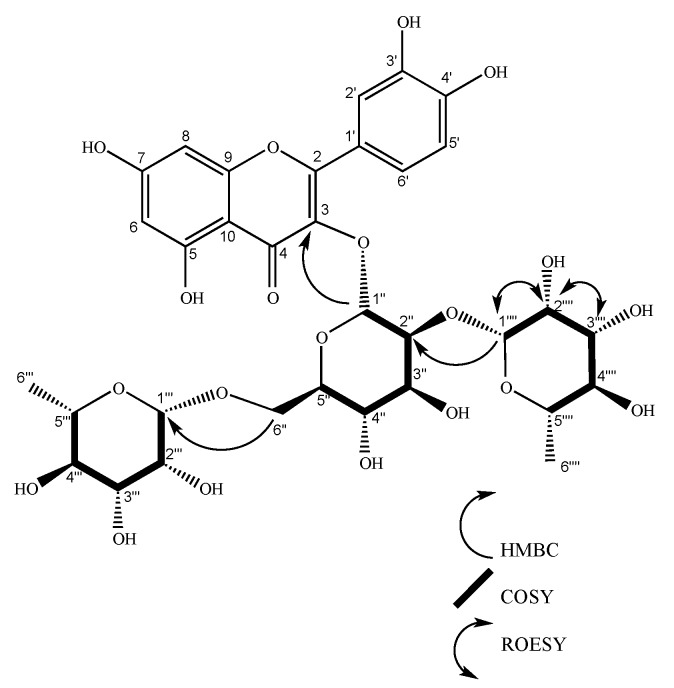
Structure of puchikrin. COSY and HMBC key correlations are shown for glycosydic units.

**Figure 8 molecules-24-04267-f008:**
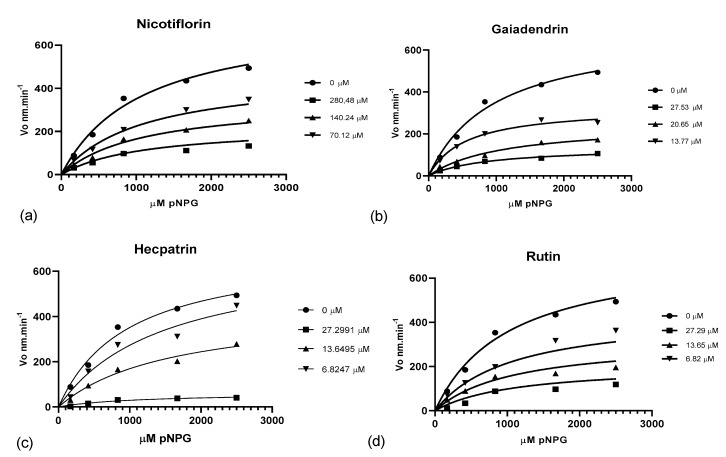
Plot of glycoside flavonoids nicotiflorin (**a**), gaiadendrin (**b**), hecpatrin (**c**), and rutin (**d**), showing variation in Vmax with three concentrations of inhibitor.

**Table 1 molecules-24-04267-t001:** Spectral NMR information of hecpatrin, gaiadendrin, and puchikrin.

	Hecpatrin (4)					Gaiadendrin (5)					Puchikrin (6)				
C	*^13^C*	*^1^H*	*COSY*	HMBC	NOESY	*^13^C*	*^1^H*	*COSY*	HMBC	ROESY	*^13^C*	*^1^H*	*COSY*	HMBC	ROESY
4	179.33					179.25			8		177.84			8	
7	166.05					165.84			6, 8		164.17			6, 8	
5	163.19					163.11			6		161.71			6	
9	158.41			2’		158.89			6’, 2’		157.49			6’	
2	158.36					158.44			8		157			8	
4’	149.58			6’’, 2’		149.54			6’, 2’, 5’		148.11			5’, 2’, 6’	
3’	146.03					145.91			6’, 2’, 5’		144.48			5’, 2’, 6’	
3	134.53					134.42			1’’		133.01			1´´	
1’	123.47			5’		123.52			6’, 2’, 5’		122.08			5’, 2’, 6’	
6’	123.21	7.60 (d, J = 2.2 Hz, 1H)	5’	2’		123.44	7.62 – 7.59 (m, 2H),	5’	5’		122.01	7.64 – 7.57 (m, 2H),	5’	5’	
2’	117.16	7.62 (d, J = 0.8 Hz, 1H)		6’		117.39					115.96				
5’	115.97	6.87 (d, J = 8.7 Hz, 1H)	6’	2’		116.04	6.87 (dd, J = 8.0, 0.7 Hz, 1H),	2’, 6’	6’, 2’, 5’		114.6	6.87 (d, *J* = 8.2 Hz, 1H)	2’, 6’	2’, 6’	
10	105.84			8		105.83			6, 8		104.46			6, 8	
1’’’’						102.63	5.23 (d, J = 1.6 Hz, 1H),	2’’’’	2’’, 5’’’’	5’’’’, 2’’	101.21	5.22 (d, *J* = 1.7 Hz, 1H)	2’’’’	2’’	2’’’’
1’’’	102.65	5.23 (d, J = 1.6 Hz, 1H)	2’’’	2’’	2’’’, 2’’	102.24	4.51 (d, J = 1.6 Hz, 1H),	2’’’	6´’, 5’’’	5’’’, 6’’	100.81	4.51 (d, *J* = 1.7 Hz, 1H)	2’’’	5’’’, 6´´	
1’’	100.31	5.75 (d, J = 7.7 Hz, 1H)	2’’	2’’	5’’	100.47	5.59 (d, J = 7.6 Hz, 1H),	2’’	3, 2’’		99.04	5.59 (d, *J* = 7.7 Hz, 1H)	2´´	3, 2’’	
6	99.79	6.18 (d, J = 2.1 Hz, 1H)		8		99.81	6.18 (d, J = 2.1 Hz, 1H),		8		98.29	6.19 (d, *J* = 2.1 Hz, 1H)		8	
8	94.58	6.37 (d, J = 2.1 Hz, 1H)		6		94.73	6.37 (d, J = 2.1 Hz, 1H),		6		93.24	6.37 (d, *J* = 2.1 Hz, 1H)		6	
2’’	80.1	3.66 (dd, J = 7.7, 9.14 Hz, 1H)	1’’	1’’’, 5’’		80.02	3.65 (dd, J = 9.2, 7.6 Hz, 1H),	1’’, 3’’	1’’’’, 3’’		78.59	3.64 (dd, J = 9.2, 7.6 Hz, 1H)	1´´,3´´	1´´´´	
4’’’	74.05	3.34 (d, J = 1.5 Hz, 1H),	3’’’	6’’’		78.9	3.56 (d, J = 8.7 Hz, 1H),				77.47	3.55 (d, J = 8.7 Hz, 1H)	4’’’’		
4’’	78.35	3.26 – 3.19 (m, 1H)	3’’, 5’’			77.06	3.33 (q, J = 2.4, 1.9 Hz, 1H)		5’’,		75.65	3.34 – 3.32 (m, 1H)		6’’, 5’’	
4’’’’						74.04	3.36 (d, J = 3.7 Hz, 1H)	5’’’’, 3’’’’	2’’’’, 6’’’’	5’’’’	72.61	3.39 – 3.35 (m, 1H)	5’’’’, 3’’’’	6’’’’	
3’’’	72.29	3.76 (t, J = 3.2 Hz, 1H)	4’’’, 2’’’			73.86	3.26 – 3.21 (m, 1H)		5’’’, 2’’’, 6’’’		72.42	3.24 (d, J = 6.6 Hz, 1H)	3’’’,	6’’’	
2’’’’						72.39	4.01 (dd, J = 3.4, 1.7 Hz, 1H),	1’’’’, 3’’’’	1’’’’	5’’’’	70.96	4.01 (dd, J = 3.4, 1.7 Hz, 1H)	1’’’’, 3’’’’	1’’’’	3’’’’, 1’’’’
3’’’’						72.28	3.81 (d, J = 3.4 Hz, 1H)	2’’’’	2’’’’		70.85	3.78 (d, J = 3.4 Hz, 1H)	4’’’’, 2’’’’	1’’’’, 2’’’’	
3’’	71.69	3.36 (t, J = 9.3 Hz, 1H)	4’’			72.24	3.50 (dd, J = 9.5, 3.4 Hz, 1H)	5’’, 2’’			70.81	3.49 (dd, J = 9.5, 3.4 Hz, 1H)		2’’	
2’’’	72.4	4.00 (dd, J = 3.4, 1.7 Hz, 1H)	1’’’, 3’’’	1’’’		72.12	3.59 (dd, J = 3.4, 1.6 Hz, 1H)	1’’’,	1’’’		70.7	3.59 (dd, J = 3.4, 1.7 Hz, 1H)			
5’’	78.95	3.56 (td, J = 8.9, 1.7 Hz, 1H)	4’’, 6’’	2’’		71.85	3.29 – 3.26 (m, 1H)	3’’	6’’, 3’’		70.42	3.30 – 3.26 (m, 1H)		3’’	
5’’’’						69.95	4.09 (dd, J = 9.7, 6.2 Hz, 1H)	1’’’’,4’’’’, 6 ’’’’	6’’’’, 1’’’’, 4’’’’	2’’’’, 4’’’’	68.51	4.08 (dd, J = 9.6, 6.2 Hz, 1H)	6’’’’, 4’’’’	1’’’’, 4’’’’, 6’’’’	1’’’’, 2’’’’
5’’’	69.96	4.07 – 4.02 (m, 1H)	6’’’	6’’’, 1’’’		69.71	3.41 (d, J = 6.2 Hz, 1H)	6’’’	6’’’, 1’’’, 3’’’,2’’’		68.28	3.41 (d, J = 6.2 Hz, 1H)	6’’’	6’’’, 1’’’	
6’’	62.55	3.72 (d, J = 2.2 Hz, 1H), 3.55 (m, 1H)	5’’	4’		68.26	3.84 (d, J = 1.4 Hz, 1H), 3.39 (d, J = 6.5 Hz, 1H)		1’’’, 4’’, 5’’		66.84	3.83 (d, J = 1.5 Hz, 1H), 3.39 (m, 1H)		1´´´	
6’’’	17.46	0.97 (d, J = 6.2 Hz, 3H)	5’’’	5’’’		17.81	1.08 (d, J = 6.2 Hz, 3H)	5’’’, 4’’’’	5’’’	2’’’	16,37	1.08 (d, *J* = 6.2 Hz, 3H)	5’’’	3’’’	
6’’’’						17.5	1.01 (d, J = 6.2 Hz, 3H)	5’’’’	5’’’’	4’’’’, 5’’’’	16.07	1.00 (d, *J* = 6.2 Hz, 3H)	5’’’’	4’’’’	

Note: All spectra were measured in methanol-d4 with the exception of ROESY of gaiadendrin and puchikrin (D_2_O).

**Table 2 molecules-24-04267-t002:** Minimum inhibitory concentration (MIC) values of the hydroalcoholic extracts of leaves and flowers, and of the five different isolated compounds against five different strains of microorganism.

Sample	*Micrococcus luteus*	*Candida albicans*	*Staphylococcus aureus*	*Escherichia coli*	*Enterococcus faecalis*
Leaf extract	500	>4000	1000	>4000	1000
Flower extract	2000	>4000	>4000	>4000	>4000
Hecpatrin	>327.59	>327.59	>327.59	>327.59	>327.59
Rutin	>327.59	>327.59	>327.59	>327.59	>327.59
Gaiadendrin	>264.32	>264.32	>264.32	>264.32	>264.32
Puchikrin	>264.32	>264.32	>264.32	>264.32	>264.32
Nicotiflorin	>336.41	>336.41	>336.41	>336.41	>336.41
Gentamicin (1000 μg/mL)	<0.39	-		0.78	125
Penicillin (250 μg/mL)	12.5	-	<0.097	-	6.25
Amphotericin B (200 μg/mL)	-	<0.075		-	-

MIC values are expressed as μg/mL for extracts and positive controls, and μM for pure isolated compounds.

**Table 3 molecules-24-04267-t003:** Results of the α-glucosidase inhibition activity of the hydroalcoholic extracts of the leaves and flowers and of the five different isolated compounds.

No.	Sample	α-Glucosidase IC_50_ (SE) (mg/mL, µM†)
**1**	Leaves extract	2.13 (0.07)
**2**	Flowers extract	51.89 (4.49)
**3**	Hecpatrin	9.66 (1.04) †
**4**	Rutin	7.53 (1.08) †
**5**	Gaiadendrin	16.87 (1.02) †
**6**	Puchikrin	NA †*
**7**	Nicotiflorin	159.09 (9.54) †
**8**	Acarbose	300–700 †

* Not active at the maximum dose tested (220.26 µM). SE: standard error. † IC50 in µM units

**Table 4 molecules-24-04267-t004:** Kinetics results of the inhibitory activity of isolated compounds against α-glucosidase enzyme.

Compound	Dose (μM)	V. max (SE) (nM.min^−1^)	Km (μM)	Inhibition Type	Ki
**No inhibitor**	0	697.7 (67.26)	985.6 (226.5)		
**Hecpatrin**	27.3	62.0 (13.41)	1107.0 (545.8)	C	3.92 (1.48)
	13.65	412.0 (159.3)	1587.0 (573.0)		
	6.82	680.0 (159.1)	1492.0 (710.0)		
**Rutin**	27.3	184.6 (47.97)	1351.0 (741.6)	NC	10.53 (1.07)
	13.65	242.7 (20.23)	624.9 (148.1)		
	6.82	583.0 (46.54)	1489.0 (241.6)		
**Gaiadendrin**	27.53	138.2 (11.40)	881.5 (11.40)	NC	12.29(1.58)
	20.65	250.1 (30.26)	1099.0 (30.26)		
	13.77	328.7 (24.77)	676.4 (8.86)		
**Nicotiflorin**	280.48	170.3 (14.63)	756.2 (171.5)	NC	124.6 (8.65)
	140.24	379.0 (44.98)	1297.0 (330.3)		
	70.12	540.8 (29.47)	1360.0 (156.2)		

SE: standard error; C: competitive inhibition; NC: non-competitive inhibition.

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
