# Peer review of "Novel Flavonoid Glycosides of Quercetin from Leaves and Flowers of Gaiadendron punctatum G.Don. (Violeta de Campo), used by the Saraguro Community in Southern Ecuador, Inhibit α-Glucosidase Enzyme"

_molecules, 2019, doi:10.3390/molecules24234267_

Round 1
Reviewer 1 Report
The manuscript needs deep revision before it could be accepted for publication. The following points should be considered.
Revise abstract and introduction to make it more informative. Have an English correction. Page 2, line 42: What is Saraguro community? Did you get the exact mass of the compounds, especially your claimed new? Page 3, line 71: Compute the degrees of unsaturation and explain in the text. I suggest to run the new compound to HR-MS. This is subject to question. There are some matches in molecular weight. Please check Kaemferol-3-o-gentiobioside-7-o-rhamnoside. Also check C-NMR and compare related match. You need to establish the identity of the compound. Lines 71 and 72: How did you compute [M+H+Na]? I suggest you get the exact mass using high res MS. This is very important for new compounds. Fig 2. Did you see HMBC connectivities in closing the rings of the glycosides? I suggest to check ROESY and NOESY data also. You need connectivities here to prove the ring structures. Get 1H-NMR integration of gaidendrin and reflect it on Figure S5 and the number of hydrogens should be reflected on the tables. Figure S9. You just choose some cross peaks to be labelled. Some are not labelled. It will affect analysis. Better explain what are those cross peaks. Figure S10. Some cross peaks are not labelled and accounted for. You need to explain what are those cross peaks. Check how did you calculate the molecular weight. Line 88 and other compounds: g/mol is not appropriate for molecular weight here. Check other papers how to present this, Da or amu. Check degrees of unsaturation, and discuss. Fig 3. Establish complete HMBC and H-H connectivities with all your structures. Be consistent with your discussions on structure elucidation. Explain lines 107 to 112 in terms of HMBC and H-H COSY than just merely writing the values. This is applicable to all compounds. Line 263-265. Put all details of the materials and solvents used. Make one figure summary of all your isolated compounds. Expand discussion. So weak. Put makers and company in your purchased solvents and materials In 3.4, what does fragmentation means? Confusing. I suggest to run your compounds in HR MS. It is very important in structure elucidation.
Author Response
Revise abstract and introduction to make it more informative.The abstract and the introduction were adjusted, in order to give more information about the performed research. We included the analytical approach used for confirming the absolute stereochemistry of carbohydrates.
Have an English correction.The entire text has been reviewed and edited, especially in English grammar.
Page 2, line 42: What is Saraguro community?A clarification about this point is now in the introduction text:
Saraguro community is one of the Kichwa indigenous groups of Ecuador, that has settled from centuries in the Loja Province, southern Ecuador. This indigenous community has an ancestral organization and preserves their traditional knowledge, medicine and culture. From ancestral times, the ethnic groups of Ecuador, as the Saraguros, are known by the use of numerous medicinal plants, and by managing their own health care system.
Did you get the exact mass of the compounds, especially your claimed new?
The exact mass of new compounds was calculated according with its molecular formula. In two cases we measured LC-MS-ESI mass and it
Page 3, line 71: Compute the degrees of unsaturation and explain in the text.
The information was included and discussed in the results of every new compound
I suggest to run the new compound to HR-MS.
We don’t have access to HR-MS
This is subject to question. There are some matches in molecular weight. Please check Kaemferol-3-o-gentiobioside-7-o-rhamnoside. Also check C-NMR and compare related match. You need to establish the identity of the compound.
Kaempferol 3-beta-gentiobioside is a kaempferol O-glucoside in which the hydroxy hydrogen at position 3 of Kaempferol has been replaced by a gentiobiosyl group. It has a role as a Brassica napus metabolite. It is a disaccharide derivative, a kaempferol O-glucoside and a trihydroxyflavone. It derives from a gentobiose. We have checked the NMR analysis and specially the HMBC experiment in our samples doesn’t correspond to the structure suggested.
Lines 71 and 72: How did you compute [M+H+Na]? I suggest you get the exact mass using high res MS. This is very important for new compounds.We included the LC/MS-ESI analysis for confirmation of the molecular mass of two new compounds. Normally at using ESI analysis some more protons as Na+, K+, H+. In our case we use this information only for confirmation. The absolute configuration of the structure is based in 1D-2D-NMR analysis and hydrolysis-derivatization-GC/MS analysis and comparation with known samples of carbohydrates, in the case of mannose and glucos.
Fig 2. Did you see HMBC connectivities in closing the rings of the glycosides? I suggest to check ROESY and NOESY data also. You need connectivities here to prove the ring structures.ROESY and NOESY analysis were included and the analysis that confirm connectivities and structure when stablished for the analysis too. We used COSY and Selective TOCSY for the assignation of proton multiplicity in sugars.
Get 1H-NMR integration of gaidendrin and reflect it on Figure S5 and the number of hydrogens should be reflected on the tables.The complete information now is reflected on the corresponding tables, and in the corresponding Supplementary material. The new code for this spectra is S36. The integration is present in the spectra at S36 and table 2 reflects the hydrogens present.
Figure S9. You just choose some cross peaks to be labelled. Some are not labelled. It will affect analysis. Better explain what are those cross peaks.HSQC analysis confirms the C-H bonding in the molecule, in this case, gaiadendrin. The new code for this spectra is S40. You can observe a zoom that clarifies the relationship between the different C-H.
Figure S10. Some cross peaks are not labelled and accounted for. You need to explain what are those cross peaks. Check how did you calculate the molecular weight.
New code for this supplementary information is S41. It is possible to observe the entire HMBC analysis. This information was also included in the corresponding table 2. In the case of the figure 6 we only included the principal HMBC relationships. Molecular weight was calculated in basis to the flavone central structure and the carbohydrates present and analyzed.
Line 88 and other compounds: g/mol is not appropriate for molecular weight here.This information was amended. We used only Da.
Check other papers how to present this, Da or amu. Check degrees of unsaturation, and discuss. Fig 3.We finally present the information in Da The discussion about degrees of unsaturation was developed.
Establish complete HMBC and H-H connectivities with all your structures. Be consistent with your discussions on structure elucidation.
As gaiadendrin, we corrected and extended the information in all the structures, including COSY, HMBC and selective TOCSY that were fundamental in the elucidation of the structure.
We reorganized our supplementary material.
S22 – S33 correspond to hecpatrin
S34 – S45 correspond to gaiadendrin
S46 – S57 correspond to puchikrin
In all the cases the tables presented in the text were completed and the discussion enhanced.
In 3.4, what does fragmentation means? Confusing.
It was a mistake, it meant fractionation. This was amended in the text.
I suggest to run your compounds in HR MS. It is very important in structure elucidation.
As stated, at present we don’t have access to HR-MS analysis in our country. However we think that the entire analysis performed confirm the absolute configuration of the new molecules. Moreover, some additional information for nicotiflorin and artabotryside A will be useful for future comparisons.
Reviewer 2 Report
The manuscript has novel information about a Loranthaceae species from Ecuador. The experiments were well planned and their results correctly analyzed. New flavonoids were found in the plant extracts and the biological activities (antimicrobial, antioxidant) were proved. The conclusions are adequate. It is very well written.
Author Response
The manuscript has novel information about a Loranthaceae species from Ecuador. The experiments were well planned and their results correctly analyzed. New flavonoids were found in the plant extracts and the biological activities (antimicrobial, antioxidant) were proved. The conclusions are adequate. It is very well written.
No comments. Thank you very much for your repply.
Reviewer 3 Report
Lines 53, 59 and 73 need to be edited.
The compounds are numbered as carbohydrates but named as tetrahydropyran derivatives. This causes issues with the numbering. e.g. a methyl group is labeled as C6 in the figure but the IUPAC C-6 being in the ring. (Check out the name and you will see that the name gives absolute configurations to methyl groups. This is a minor issue which can be fixed by re-numbering the figure or you can leave it alone if that is the editor's desire.
My experience is that elemental analysis (aka exact mass) measurements from LC/MS are required for most journals. One should be able to get a formula to a mili mass unit. The data here only reports to the tenth.
The authors never stated how they determined the absolute stereochemistry of the carbohydrates. This is normally done by hydrolyzing the glycoside and analyzing the glycosides via GC/MS against known standards. Perhaps this journal does require this much rigor.
Author Response
1. Lines 53, 59 and 73 need to be edited.
The entire text has been reviewed and edited, especially in English grammar.
2. The compounds are numbered as carbohydrates but named as tetrahydropyran derivatives. This causes issues with the numbering. e.g. a methyl group is labeled as C6 in the figure but the IUPAC C-6 being in the ring. (Check out the name and you will see that the name gives absolute configurations to methyl groups. This is a minor issue which can be fixed by re-numbering the figure or you can leave it alone if that is the editor's desire.
As presented in other related papers with glycosylated flavones we decided to give names related to the central flavone structure and the sugars linked. For this reason we revised and changed the names.
3. My experience is that elemental analysis (aka exact mass) measurements from LC/MS are required for most journals. One should be able to get a formula to a mili mass unit. The data here only reports to the tenth.
The exact mass of new compounds was calculated according with its molecular formula. Our LC/MS system gets only to the hundredth, as reported, and that information was used only for confirmation where possible. The absolute stereochemistry was obtained by 1D-2D NMR experiments (1H, 13C, COSY, selective TOCSY, HSQC, HMBC), bibliographic information and hydrolysis of glycosides and GC/MS analysis.
The authors never stated how they determined the absolute stereochemistry of the carbohydrates. This is normally done by hydrolyzing the glycoside and analyzing the glycosides via GC/MS against known standards. Perhaps this journal does require this much rigor.We added to the paper the confirmation of the presence of the carbohydrates by hydrolyzing the glycoside in the new compounds and analyzing through GC-MS using standards. The 1D-2D NMR studies (specially COSY, HMBC, TOCSY) and precedent studies confirm the absolute stereochemistry
Round 2
Reviewer 1 Report
This is a great improvement for the manuscript. I recommend to get the stereochemistry of the sugars using NOESY and ROESy data. Also, the authors could tabulate all structure elucidation data of all compounds in just one table for quick reference.
Author Response
This is a great improvement for the manuscript. I recommend to get the stereochemistry of the sugars using NOESY and ROESy data. Also, the authors could tabulate all structure elucidation data of all compounds in just one table for quick reference.
Answer:
Thank you very much for your reviews.
We accepted the recommendations and analyzed and conclude the absolute stereochemistry using NOESY (for Hecpatrin) and ROESY (Gaiadendrin and Puchikrin).
We tabulated the data of the new compounds in only one table.
Best regards,
The authors.